# Utility of Gene Panels for the Diagnosis of Inborn Errors of Metabolism in a Metabolic Reference Center

**DOI:** 10.3390/genes12081262

**Published:** 2021-08-19

**Authors:** Sofia Barbosa-Gouveia, María E. Vázquez-Mosquera, Emiliano González-Vioque, José V. Álvarez, Roi Chans, Francisco Laranjeira, Esmeralda Martins, Ana Cristina Ferreira, Alejandro Avila-Alvarez, María L. Couce

**Affiliations:** 1Unit of Diagnosis and Treatment of Congenital Metabolic Diseases, Department of Paediatrics, IDIS-Health Research Institute of Santiago de Compostela, Centro de Investigación Biomédica en Red de Enfermedades Raras (CIBERER), European Reference Network for Hereditary Metabolic Disorders (MetabERN), Santiago de Compostela University Clinical Hospital, 15704 Santiago de Compostela, Spain; sofiabsg@gmail.com (S.B.-G.); maria.eugenia.vazquez.mosquera@sergas.es (M.E.V.-M.); jvictor_alvarez@hotmail.com (J.V.Á.); roi.chans.gerpe@sergas.es (R.C.); 2Department of Clinical Biochemistry, Puerta de Hierro-Majadahonda University Hospital, 28222 Majadahonda, Spain; egvioque@gmail.com; 3Biochemical Genetics Unit, Centro de Genética Médica Doutor Jacinto Magalhães, 4050-466 Porto, Portugal; francisco.laranjeira@chporto.min-saude.pt; 4Centro Materno-Infantil do Norte, Centro Hospitalar Universitário do Porto (CHUP), Coordinator of the Centro de Referência de Doenças Hereditárias do Metabolismo do CHUP, 4050-466 Porto, Portugal; esmeralda.g.martins@gmail.com; 5Hospital D. Estefânia, Centro Hospitalar de Lisboa Central (CHLC), Coordinator of the Centro de Referência de Doenças Hereditárias do Metabolismo do CHLC, 1169-050 Lisboa, Portugal; anacristina.ferreira@chlc.min-saude.pt; 6Neonatology Unit, Pediatrics Department, Complexo Hospitalario Universitario de A Coruña, SERGAS, 15006 A Coruña, Spain; alejandro.avila.alvarez@sergas.es

**Keywords:** differential diagnosis, genetic diagnosis, inborn errors of metabolism

## Abstract

Next-generation sequencing (NGS) technologies have been proposed as a first-line test for the diagnosis of inborn errors of metabolism (IEM), a group of genetically heterogeneous disorders with overlapping or nonspecific phenotypes. Over a 3-year period, we prospectively analyzed 311 pediatric patients with a suspected IEM using four targeted gene panels. The rate of positive diagnosis was 61.86% for intermediary metabolism defects, 32.84% for complex molecular defects, 19% for hypoglycemic/hyperglycemic events, and 17% for mitochondrial diseases, and a conclusive molecular diagnosis was established in 2–4 weeks. Forty-one patients for whom negative results were obtained with the mitochondrial diseases panel underwent subsequent analyses using the NeuroSeq panel, which groups all genes from the individual panels together with genes associated with neurological disorders (1870 genes in total). This achieved a diagnostic rate of 32%. We next evaluated the utility of a tool, Phenomizer, for differential diagnosis, and established a correlation between phenotype and molecular findings in 39.3% of patients. Finally, we evaluated the mutational architecture of the genes analyzed by determining z-scores, loss-of-function observed/expected upper bound fraction (LOEUF), and haploinsufficiency (HI) scores. In summary, targeted gene panels for specific groups of IEMs enabled rapid and effective diagnosis, which is critical for the therapeutic management of IEM patients.

## 1. Introduction

Recent advances in precision medicine and next-generation sequencing (NGS) technologies are helping to uncover the molecular bases of inborn errors of metabolism (IEM). These genetic disorders, although rare at the individual level, are collectively common, with an estimated worldwide incidence of 1 per 1900 births [1]. IEMs are caused by defects in enzymes, cofactors, and transporters in the biochemical pathways that metabolize proteins, lipids, carbohydrates, vitamins, nitrogenous compounds, and others, resulting in the accumulation of toxic substrates or deficiencies in essential metabolites [2]. These defects in turn give rise to heterogeneous clinical presentations and chronic problems such as neurodevelopmental deficits and acute metabolic decompensation [3]. The age of onset can range from childhood to adolescence and even adulthood, although the most severe forms tend to manifest during the neonatal period and are associated with significant mortality [4].

Evidence supporting genotype–phenotype correlations in IEMs is not always clear [5]. The challenges in diagnosing these disorders are largely attributable to marked clinical and genetic heterogeneity, the diversity of metabolic pathways involved, and a lack of clinical awareness of these rare diseases [1]. Delays or difficulties in establishing a definitive diagnosis are common, and many of these patients remain undiagnosed [6]. However, early diagnosis of IEMs is crucial, especially in cases of disorders that are treatable and can be managed with dietary and supportive therapy or other approaches such as enzyme replacement or gene therapy [7].

Rates of IEM diagnosis have improved in recent decades with the advent of specific tiered testing. First-tier laboratory tests can be incorporated into newborn screening (NBS) programs that screen neonates in a given population for life-threatening or long-term health conditions in order to reduce morbidity and mortality [8]. IEMs included in these NBS programs are selected based on specific criteria first developed by Wilson and Junger in 1968 [9], and reevaluated in 2006 by Ross to include disorders that do not fulfil the initially approved criteria [10]. However, the first-tier laboratory tests can also be used when a patient presents a clinical phenotype suggestive of an IEM, such as an acute metabolic presentation, hepatomegaly, liver and kidney involvement, muscle weakness, or intellectual impairment. Using this approach, previously identified metabolites can be analyzed through tandem mass spectrometry (MS-MS) and linked to a given metabolic disorder [11]. 

Diagnosis of IEMs remains challenging as there are many metabolic disorders for which no specific biomarker has been identified or that present with a broad spectrum of clinical signs that can complicate diagnosis. To address these challenges, molecular testing strategies have been applied to both the molecular study and genetic diagnosis of IEMs. NGS technologies are largely responsible for the significant improvements in the diagnosis of these disorders in clinical practice [12]. Thus, when there is a strong clinical suspicion of inherited metabolic disease (IMD) but the patient presents with highly heterogeneous clinical signs or there are insufficient biochemical findings to guide a specific diagnosis, options include NGS using gene panels specifically designed for a group of molecularly well-characterized disorders, clinical exome sequencing (CES), or whole-exome sequencing (WES) [5,13,14].

It is well established that alterations in different functional regions of a given gene can affect protein domains or catalytic regions of enzymes, thereby impacting the function of multiple metabolic pathways and giving rise to a variety of potential phenotypes [5,15]. While NGS technologies have been helpful in furthering our understanding of the molecular bases of IEMs, it is critical to understand the mutational burden within a given gene in order to determine its tolerance to mutation [16]. This enables assessment of missense and loss-of-function (LOF) rare variants in the general population and prioritization of variants in highly penetrant genes that are intolerant to mutation [17,18]. This way, the pathogenic significance of a given rare protein-altering variant can be inferred, allowing accurate interpretation of the variant and association with the clinical phenotype.

In this study, we evaluate the mutational architecture to understand and prioritize variants in genes associated with IEMs, assess the utility of specific gene panels for the diagnosis of metabolic disorders, and discuss the implications of a rapid positive molecular diagnosis. Despite the availability of NGS, there are several IEMs that we identified for which molecular data are lacking. It is therefore important to harness the advances made in genetic NBS programs in order to identify pathogenic variants and add new genes to the panels used in these screening programs, ultimately improving the capabilities of preventive medicine.

## 2. Materials and Methods

### 2.1. Study Design

This prospective multicenter study was conducted over a 3-year period in the Metabolic Unit of the University Clinical Hospital of Santiago de Compostela (Spain). Patients included were from 20 centers in Spain and Portugal. In all cases, before genetic analysis was performed informed consent was obtained from the patient’s legal guardians. This study was approved by the Clinical Research Ethics Committee of Galicia (reference code 2015/410).

### 2.2. Study Population

Patients aged 0–18 years with a suspected IMD were included in this study. For each patient we evaluated the following variables: sex, age of symptom onset, clinical signs and symptoms, laboratory test parameters, family history, consanguinity data, genetic analysis of patients and parents, and treatments administered (nutritional and/or pharmacological). The relevant clinical information, including clinical symptoms and complementary data, were collected using Human Phenotype Ontology (HPO) terms (https://hpo.jax.org/, accessed on 10 May 2021) [19]. All HPO terms for each patient with a positive diagnosis are shown in Appendix A.

For confirmation of disease segregation, family studies were performed whenever possible. Analysis of HPO terms using the Phenomizer tool allowed us to correlate the phenotype with the obtained molecular diagnosis (compbio.charite.de/phenomizer/, accessed on 10 May 2021) [20]. This computer-based tool can facilitate the diagnostic process by combining specific phenotypic features, correlating them with a genetic disorder, and generating a corresponding *p*-value.

### 2.3. Gene Panel Design

We designed four different multi-gene panels consisting of a group of genes previously described in the literature and associated with inborn errors of intermediary metabolism, hypoglycemic/hyperglycemic events associated with metabolic disorders or other processes, mitochondrial diseases, and complex molecular defects, including leukodystrophies. These multi-gene panels were updated throughout the 3-year study period by reviewing the most recent scientific publications to include new genes potentially associated with IEMs. Several versions of each panel were designed and are available upon request. The genes included in the latest version of each gene panel are shown in Appendix A. During the course of the study, we became aware of the difficulties encountered by clinicians in choosing a specific gene panel when the clinician is facing a complex clinical case. For this reason, we have clustered together all the genes from the individual panels, along with genes associated with neurological disorders, to design the NeuroSeq panel. The NeuroSeq was used in patients with suspicion of mitochondrial disorders with a negative result in the individual mitochondrial panel. A description for each genetic panel is provided below:-Inborn errors of intermediary metabolism (INT MET) panel: this panel included genes associated with small molecule diseases linked to an accumulation of compounds causing acute or progressive “intoxication” disorders, and small molecule diseases linked to a deficiency, including all the defects of essential molecules that must be transported across cell membranes. The first designed version included 138 genes; the last panel version included 172 genes.-Hypoglycemic/ hyperglycemic events (HYPO/HYPER) panel was associated with congenital metabolic disorders or other processes: this panel included genes associated with familial hyperinsulinism, monogenic diabetes (neonatal, MODY), and other disorders in which hypoglycemic/hyperglycemic events are a predominant sign, (i.e., Alström Syndrome, Shashi-Pena syndrome, and tubulointerstitial kidney disease). The first designed version included 22 genes; the last panel version included 65 genes.-Mitochondrial diseases (MITO) panel: this panel included nuclear genes coding for respiratory chain complex subunits, proteins involved in the oxidative phosphorylation system (OXPHOS) function, or candidate genes. The first designed version included 176 genes; the last panel version included 320 genes.-Deficiency of complex molecules, including leukodystrophies (COMP MOL) panel: this panel included genes associated with complex molecule disorders that involve sphingolipids, phospholipids, cholesterol and bile acids, glycosaminoglycans, oligosaccharides, glycolipids, and nucleic acids. The first designed version included 141 genes; the last panel version included 177 genes.-NeuroSeq panel: this panel included 1870 genes associated with metabolic disorders described previously and genes linked to neurologic disorders such as epilepsy, intellectual disability, cerebral morphogenesis defects, and neuromuscular and ataxia disorders (except those caused by repeat expansion).

### 2.4. Genetic Analysis

The genetic data were analyzed through NGS technologies consistent in enrichment with an in-solution hybridization technology (Sure Select XT; Agilent Technologies, Santa Clara, CA, USA), followed by sequencing using the Miseq platform (Illumina, San Diego, CA, USA) for the individual gene panels and the NextSeq platform for the NeuroSeq panel. A custom Sure Select probe library was designed to capture the exons and exon-intron boundaries of the targeted genes [21]. Sequence capture, enrichment, and elution were performed according to the manufacturer’s instructions. Image analysis and processing of the fluorescence intensities in sequences (“base calling”) was performed with Real Time Analysis (RTA) software v.1.8.70 (Illumina), and the FastQC v0.10.1 program was used for data quality control. Reads were aligned to the reference genome GRCh37 with BWA v0.7.9a [22]. NGSrich v0.7.5 software [23] was used as a control prior to variant detection, and BEDTools 2.17.0 [24] and Picard 1.114 [25] for intermediate steps. VarScan v.2.3.6 [26] and SAMtools v0.1.19 [27] variant detection software were used for indels and SNP, respectively, and Annovar for variant annotation [28]. 

To ensure a reliable clinical interpretation of the variants detected, we applied prioritization criteria to predict pathogenicity according to ACMG guidelines [29]. Based on this classification system, variants are assigned to one of five pathogenicity classes: likely benign, benign, uncertain significance, likely pathogenic, and pathogenic. To identify and prioritize genes sensitive to mutational changes (i.e., those most likely to contribute to disease), z-scores and loss-of-function observed/expected upper bound fraction (LOEUF) were analyzed using the Genome Aggregation Database (gnomAD)—Predicted Constraint Metrics [30], as well as haploinsufficiency (HI) scores from ClinGen data [31]. These measures were used to determine how intolerant a gene can be to a specific type of variant. Intolerant genes are those that are more likely to cause disorders than genes that can tolerate functional variation. For missense variants in dominant genes, a positive z-score indicates greater intolerance to functional variation, while a negative z-score indicates less constraint (i.e., more observed variants than expected). To infer mutational tolerance in recessive and X-linked genes, we used LOEUF scores and HI scores. Low LOEUF scores (<0.35) are associated with strong selection against predicted loss-of-function (pLoF) variants in a given gene, which means that the gene cannot tolerate protein-truncating variants such as nonsense, frameshift, splice acceptor, and splice donor variants, while high LOEUF scores imply a higher tolerance to inactivation [32,33]. HI scores are used to assess the dosage sensitivity in a given gene [34].

### 2.5. Statistical Analysis

The Phenomizer algorithm was used to compare the clinical features of each patient against those of a set of annotated diseases, ranked according to *p*-values. A Chi-squared test was applied to detect differences in the diagnostic yields of the four individual gene panels, followed by Fisher’s test to detect significant differences between panels. *p*-values were adjusted for multiple testing using the Benjamini–Hochberg method [35]. A *p*-value < 0.05 was considered statistically significant.

## 3. Results

In total, we studied 311 patients: 123 (39.9%) females and 188 (60.1%) males, with a mean age of 6.71 (±5.73) and 7.18 (±5.55) years, respectively. The median coverage achieved in the samples studied and the target coverage at 20X of the genes included in the gene panels are depicted in Table 1. Median coverage ranged from 291X to 413X, corresponding to the NeuroSeq and HYPO/HYPER panels, respectively. The overall target coverage at 20X ranged from 97.29% to 99.93%, corresponding to the NeuroSeq and HYPO/HYPER panels, respectively (Table 1).

### 3.1. Diagnostic Rates

Of the 311 patients studied, 83 were diagnosed (31.83%) and 33 (12.86%) had an inconclusive diagnosis, while in 195 (55.31%) no evidence of the molecular basis of the disease was detected (Figure 1). In 53 patients (63.86%) with a clear metabolic profile we were able to achieve a confirmatory genetic analysis. Among patients for whom a diagnosis was established, the genetic result was obtained within 2–4 weeks, enabling initiation of early treatment. We evaluated the impact of genetic diagnosis on medical management (i.e., changes in medication and/or diet, redirection of patient’s care), and found that establishing a genetic diagnosis directly affected medical management in 32 patients (Appendix A), and led to initiation of palliative care in another 8 patients. Moreover, all patients and families benefited from timely genetic counseling. In 33 cases, no conclusive diagnosis was established for the following reasons: (i) in 61% of cases a pathogenic variant/VUS was detected in a gene closely related to the patient’s clinical phenotype, but the same variant was also detected in one of the parents, who was unaffected, despite a previous family history; (ii) in 2% of cases, a pathogenic variant closely related to the patient’s clinical phenotype was detected in a recessive gene, but a second variant was absent; (iii) in 37% of cases, the family studies could not be completed, and therefore we were unable to establish disease segregation (Appendix A). 

Several versions of each multi-gene panel were designed during the 3-year study period (Figure 1). An exponential increase in the rate of positive diagnosis was observed for the COM MOL panel as new genes were added. In the case of the INT MET and HYPO/HYPER panels, the rate of positive diagnosis remained relatively stable over the last few versions. 

Of all the gene panels used to study patients with suspected IEMs, the INT MET panel produced the highest diagnostic rate, achieving a definitive diagnosis in 62.86% of cases. The second highest rate of positive diagnosis (32.84%) was achieved with the COMP MOL panel, which primarily diagnosed patients with some form of lysosomal disorder. Although the molecular diagnosis of suspected mitochondrial diseases has advanced considerably in recent years, recognition of clinical phenotypes associated with these diseases remains challenging. In our cohort, the lowest rate of positive diagnosis (17%) was achieved with the mitochondrial diseases panel. We also observed significant differences in the diagnostic rate between INT MET regarding HYPO/HYPER, MITO, and COMP MOL panels (*p* < 0.0001). Moreover, significant differences in the rate of diagnosis were observed between the COMP MOL and MITO panels (*p* < 0.0126) (Figure 2).

The most common IEMs detected in this study were combined oxidative phosphorylation deficiency (11 patients), phenylketonuria (7 patients), and lysosomal disorders (7 patients).

Of the 112 patients for whom negative results were obtained using the MITO panel, 41 were reanalyzed upon request using the NeuroSeq panel. In 13 of those 41 cases (32%), we succeeded in identifying the molecular basis of the disorder and found that the gene responsible for the molecular defect was actually associated with a different disorder (usually a neurological disease), or had not yet been added to the MITO panel (Appendix A). 

We used the Phenomizer algorithm to analyze the data of 140 patients for whom more complete medical histories were available in order to identify candidate diseases based on the clinical features of each patient, and determine the corresponding *p*-values. In 55 of those cases (39.3%), we obtained a statistically significant *p*-value, based on which we could establish a correlation between the clinical features and the molecular findings (Appendix A). 

### 3.2. Types of Inheritance Identified Associated with IEMs

Although IEMs largely follow an autosomal recessive inheritance pattern, variants in autosomal dominant and X-linked genes were also identified in all individual panels (Figure 3). In cases in which the diagnosis was inconclusive, we identified a high percentage of autosomal dominant genes: in all cases a pathogenic variant or one of uncertain significance (VUS) was detected in a gene closely related to the patient’s clinical phenotype, but the same variant was also detected in one of the parents, who was unaffected, despite a previous family history.

### 3.3. Utility of z-Score, Loss-of-Function Expected Upper Bound Fraction (LOEUF), and Haploinsufficiency (HI) Score for Gene Prioritization

Comparing the expected and the observed number of rare missense variants in population databases such as gnomAD (which aggregates exomes and whole-genome data from a large group of individuals and enables identification of common and rare variants) can provide information on pathogenicity if a gene demonstrates constraint for missense variation, depending on whether that constraint is observed across the entire gene or is confined to specific regions of the gene. Larger genes are expected to have more missense variants, although this does not mean the observed variants are pathogenic. Certain genes include highly-conserved regions that are related to the strength of purifying selection and are more sensitive to variation, meaning that rare variants identified in these genes are more likely to be associated with disease. In our cohort we identified various variants in autosomal dominant and X-linked genes that are reported to be intolerant to functional variation (Figure 4a). Although there was some overlap in the genes included in each of the four individual gene panels, we detected missense variants in intolerant genes with a high z-score (z-score > 1.5 × 104) in X-linked genes (*ATP7A* and *HPRT1* in the INT MET panel; *PHKA2* in the HYPO/HYPER panel; and *DNM1L*, *PDHA1*, and *SLC6A8* in the MITO panel) and in autosomal dominant genes (*KCNJ11*, *ABCC8*, *GLUD1*, *INSR* in the HYPO/HYPER panel; and *GNE* and *SOX10* in the COMP MOL panel). We identified recessive and X-linked genes that are significantly more constrained for loss-of-function (LoF) than autosomal genes, including *PHKA2*, *PDHA1*, and *SLC16A2* (LOEUF < 0.35), which have been described previously [36,37], with sufficient evidence for haploinsufficiency (Figure 4b).

## 4. Discussion

NGS technologies show great utility, improving the rate of diagnosis of IEMs while shedding light on the complex underlying genetic architecture [12,38,39]. In this study, we detected differences in the rate of positive diagnosis between gene panels: higher rates were observed in cases for which a previous confirmatory assay was available, as for the INT MET panel, while lower rates were observed in cases of semi-untargeted analysis (e.g., the HYPO/HYPER and MITO panels). The INT MET panel showed significant differences with respect to the other three gene panels: HYPO/HYPER, MITO, and COMP MOL (*p* < 0.0001). Furthermore, we observed significant differences in the rate of positive diagnosis between the COMP MOL and MITO panels (*p* < 0.0126). Comparison of the rate of positive diagnosis across successive versions of each multi-gene panel design showed that at a certain point the addition of genes did not translate to an increase in the rate of positive diagnosis. For example, the rate of positive diagnosis remained relatively stable over the last few versions of the INT MET and HYPO/HYPER panels. There are two potential explanations for this observation. Despite recent advances in our knowledge of the molecular bases of IEMs and the discovery of disease-associated genes, (a) the analysis of NGS data from gene panels remains limited to coding regions of the genome, excluding intronic variants, meaning that we are still missing the same regions; or (b) we are failing to take into account the complex gene regulation processes that may underlie the disease, including regulation in cis by promoters, enhancers, and repressors and regulation in trans by transcription factors or microRNAs.

Patients who were analyzed using the INT MET and COMP MOL panels had a well-founded suspected diagnosis, based on their clinical history and, more importantly, biochemical analyses carried out. These two panels therefore produced the highest rates of diagnosis because the patients analyzed had biochemical findings that allowed orientation of the genetic diagnosis towards a set of well-defined genes associated with the disease [40]. In these diseases, even though in most cases the enzyme deficiency already determines the diagnosis, diagnosis should be confirmed genetically. Indeed, the genetic study is increasingly becoming the first-line diagnostic test [41]. On the one hand, patients can benefit from genetic testing to obtain clarity about their disease prognosis, prevent the development of more severe forms, and avail themselves of timely genetic counseling for their families. On the other hand, inconclusive genetic results can lead to stress and anxiety, while definitive results indicating that a patient is at increased risk of developing an incurable disease can be disappointing. In our cohort, all patients were being monitored by metabolic disease units specializing in the diagnosis and follow-up of this type of disorder, which greatly facilitated initial orientation of the suspected diagnosis.

The HYPO/HYPER panel had a lower rate of positive diagnosis (19.64%). Although hypo/hyperglycemia are common in clinical practice, the etiology of these spontaneous events remains difficult to decipher, as frequently the underlying defects do not directly influence glucose metabolism but are associated with other metabolic or non-metabolic disorders [39,42]. In these cases, the HYPO/HYPER panel is more challenging and difficult to design, and it is therefore highly likely that the causative gene may not have been included. Therefore, analysis by CES or WES is advisable to improve the rate of diagnosis when studying patients with hypo/hyperglycemia who present with clinically significant events. 

Although the molecular diagnosis of suspected mitochondrial disease has advanced significantly in recent years, recognizing clinical phenotypes associated with this disease remains a complex task [43]. In our cohort, the lowest rate of diagnosis, 17%, was achieved with the MITO panel, for which rates of 7–31% are described in the literature [44]. In 13 patients from our cohort, a mitochondrial disorder was initially suspected as the cause of the clinical presentation. However, reanalysis using the NeuroSeq panel clearly revealed the problem to be caused by the high degree of clinical heterogeneity of these disorders: in most cases, the molecular defect was actually associated with a neurological disease. Further complicating matters, phenotypic variation may be due to defects in nuclear DNA (nDNA) or in mitochondrial DNA (mtDNA), whereby the number and distribution of dysfunctional mitochondria throughout the individual’s body can vary. In the case of mtDNA defects, the phenotype will depend on the specific tissues containing abnormal mitochondria, the proportion of abnormal mitochondria within these tissues, and the number of mutant mtDNA copies within the tissue (heteroplasmy). In the case of nDNA defects, the phenotype is driven by the dependence of the tissue in question on mitochondrial respiration and the specific expression of a potentially compensatory protein or proteins in a given tissue. Analyses of interactions between both genomes should be interpreted with caution, particularly the frequency of variants. In patient 49 we detected compound heterozygous variants in *EARS2*, which is associated with combined oxidative phosphorylation deficiency. The missense variant identified in this patient was actually classified as benign according to ACMG guidelines due to its high frequency in the African population, confirmed in gnomAD, but the same variant had also been described and associated with leukoencephalopathy with thalamus and brainstem involvement and high lactate syndrome [45,46]. Furthermore, the spectrum of disease phenotypes does not always correlate with the severity of the disease-causing mutations [5]: carriers of the same variant may have differing presentations, as we observed for patient 34 and his brother, both of whom carried pathogenic variants in *FOXRED1* but presented with differing degrees of clinical severity [47]. 

Several diseases can be linked to a small number of pathogenic variants that affect the encoded protein in a specific manner. Detection of these variants is essential to not only identify the associated genes and diseases, but also to help establish a definitive diagnosis. Many of these variants are present in a given population and are directly associated with a specific disorder. These variants are usually evaluated in functional studies to confirm their pathogenicity and to classify them as the cause of the disease. In our cohort, two patients carried the missense variant c.533G > A in *HEXA*, a gene implicated in gangliosidosis. This sequence change replaces arginine with histidine at codon 178 of the *HEXA* protein (p.Arg178His). This is a recurrent variant that accounts for approximately 55% of all pathogenic alleles identified in *HEXA* [48]. Although the patients were homozygous (patient 82) and compound heterozygous (patient 63) for this variant, both presented similar symptoms and disease course. The c.533G > A variant appears to be a common cause of Tay-Sachs disease, which is more prevalent in Portugal, although it has also been identified in other countries such as Spain and Italy due to migratory flows [49,50]. In two other patients (7 and 12), the variant c.782G > A (p.Arg261Gln) in *PAH* was detected in compound heterozygosity, and was associated with hyperphenylalaninemia and classic phenylketonuria, respectively. This variant consists of an alteration in a conserved nucleotide at position 261, where the aromatic amino acid hydroxylase is located, in the C-terminal of the phenylalanine-4-hydroxylase protein. This variant has been associated with benign hyperphenylalaninemia and classic phenylketonuria requiring dietary treatment [51,52], and has an allelic frequency of 5.3–5.5% [53,54]. In vitro studies have shown that the c.782G > A variant alters the function of the protein and that the resulting enzyme retains a residual level of activity (6.3–11%), which may explain the moderate phenotype associated with the variant [55]. The identification of recurrent variants in a given population is critical so they can be accurately assessed and be included in newborn screening programs, which have been developed and optimized in the last few years [56].

Biochemical biomarkers provide a roadmap for the interpretation of variants identified in specific genes. However, such biomarkers are not always available, and there is an urgent need for new biomarkers for the identification and management of patients who, despite achieving metabolic control, can remain at risk of developing neurological, cognitive, and behavioral problems [57]. While new biomarkers are lacking, genetic diagnosis of IEMs is aided by the assessment of the patient’s phenotypic features using methods such as Phenomizer, which can facilitate differential diagnosis, and by assessment of the pathogenicity of rare variants, which are more likely to be deleterious and are expected to exert stronger-than-average effects on constrained genes (high Z-score and low LOEUF score). In this study, Phenomizer predicted a genetic diagnosis based on phenotype with statistical significance (*p* < 0.05) in 39.3% of patients for whom a complete clinical history was available. This frequency is similar to that described by Hoon Son et al. [58], in which a positive diagnosis was established in 35.7–46.4% of cases, and by Brunelli et al. [59], who achieved a rate of positive diagnosis of 40% using a targeted gene panel. We found that statistically significant data were useful to help prioritize likely pathogenic variants in genes expected to be intolerant to mutation. For example, we established diagnosis in 2 patients for whom no biochemical data were available: patient 38 (lethal encephalopathy due to defective mitochondrial peroxisomal fission) and patient 42 (pyruvate dehydrogenase deficiency). In both cases, de novo variants were detected in genes for which we obtained significant Phenomizer *p*-values, high z-scores, and high predicted dosage sensitivity. Moreover, both genes, *DNM1L* and *PDHA1*, had been previously associated with loss of function [60,61]. The identification and characterization of LoF variants that can likely lead to recessive or dominant diseases can have critical implications for diagnosis, treatment, and disease management [17]. Knowledge of the mutational burden of the genes included in the genetic analysis will enable better prioritization of variants detected by NGS sequencing. In the case of certain IEMs, the availability of biomarkers facilitates gene panel design and accurate interpretation of the genetic findings. Indeed, in patients with a clear metabolic profile, we were able to achieve a confirmatory genetic analysis. As demonstrated in the present study, targeted gene panels are highly productive and cost-effective for the diagnosis of a well-known group of disorders. We achieved a short diagnostic turnaround time of 2–4 weeks, which enabled early implementation of disease management strategies based on dietary and therapeutic options that are available for several IEMs. Establishing a genetic diagnosis led to changes in medical management or the initiation of palliative care in 40/83 (48.19%) patients.

Moreover, although WES and WGS have proven useful for the identification of novel genes, those findings are not diagnostically useful [62]: first, because functional characterization is required to determine the significance of the identified variants; and second, because these NGS strategies involve a greater turnaround time. These challenges make gene panel approaches a rapid and cost-effective method for IEM patients, enabling faster diagnosis and prompt initiation of treatment. The INT MET panel, which includes over 172 genes and offers a high diagnostic rate, is sufficiently comprehensive to cover the majority of known monogenic disorders, and can provide new insights into treatment options for defects of intermediary metabolism, as well as enabling identification of variants in family members and determination of the probability of transmission of the inherited condition to the patient’s offspring.

Despite the great advances achieved with NGS gene panels, limitations include inconclusive diagnoses and high rates of negative molecular diagnoses. In the present study, inconclusive diagnoses largely occurred in cases in which a pathogenic variant/VUS was identified in a gene closely associated with the patient’s clinical phenotype, but the same variant was identified in one of the parents, who was unaffected but had a previous family history. Such cases may be explained by incomplete penetrance or mosaicism, although further studies will be required to confirm this hypothesis. There are several potential explanations for the high rates of negative molecular diagnoses. First, the causative gene may not be included in the panel design. This underscores the importance of periodic updating of targeted gene panels. Second, certain genes may encode proteins that are closely associated with alterations of specific biomarkers but are either unknown or have not been linked with a specific disorder. Moreover, the filtering criteria applied usually exclude variants without predicted functional consequences [63]. Third, nonspecific or overlapping phenotypes can complicate orientation of the diagnosis: a given gene can be involved in multiple metabolic pathways, and therefore a pathogenic variant in that gene could give rise to different disorders with similar phenotypes. Finally, NGS is not without methodological limitations. These include difficulties in sequencing regions with repetitive sequences and in identifying epigenetic modifications and homologous genes, which hinders identification of potential pathogenic variants and elucidation of the molecular bases of IEMs [64].

## 5. Conclusions

In summary, we achieved rapid and effective diagnosis using targeted gene panel analysis, demonstrating the advantages of NGS for specific groups of IEMs. In particular, the INT MET panel provided a high diagnostic rate with a short turnaround time, which is critical to enable initiation of therapeutic management. Moreover, we showed that the use of statistical approaches to infer the mutational gene burden and phenotypic algorithm tools such as Phenomizer facilitates differential diagnosis by linking molecular findings with the clinical phenotype. Nonetheless, clinical or whole-exome sequencing should be considered for groups of disorders with high levels of clinical and biochemical heterogeneity.

## Figures and Tables

**Figure 1 genes-12-01262-f001:**
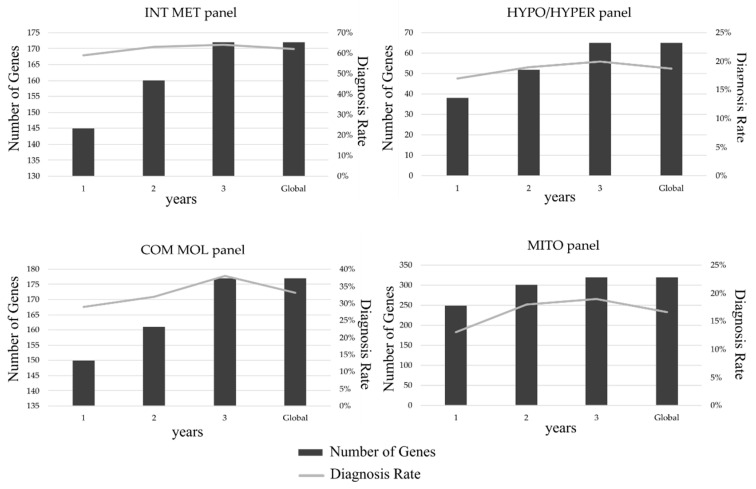
Changes in the rate of positive diagnosis (grey line) with the addition of new genes (dark bars) to each of the multi-gene panels over the 3-year study period. Global values represent the mean rate of positive diagnosis.

**Figure 2 genes-12-01262-f002:**
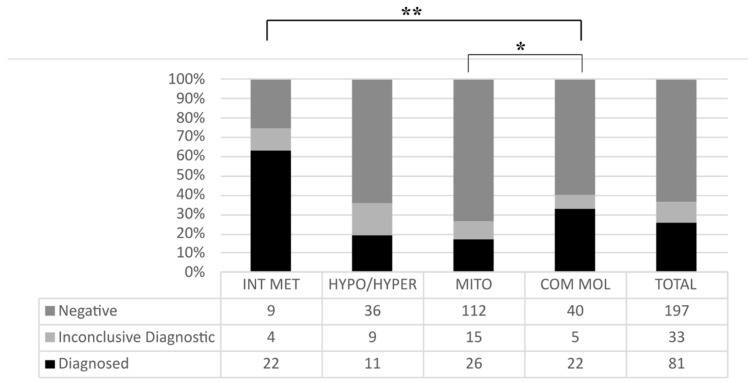
Representation of the overall rate of diagnosis achieved for each panel. For each panel the total number of patients analyzed and the corresponding diagnostic outcome are shown. * *p* < 0.05; ** *p* < 0.01 (correlational analysis with Chi-squared and Fisher’s test).

**Figure 3 genes-12-01262-f003:**
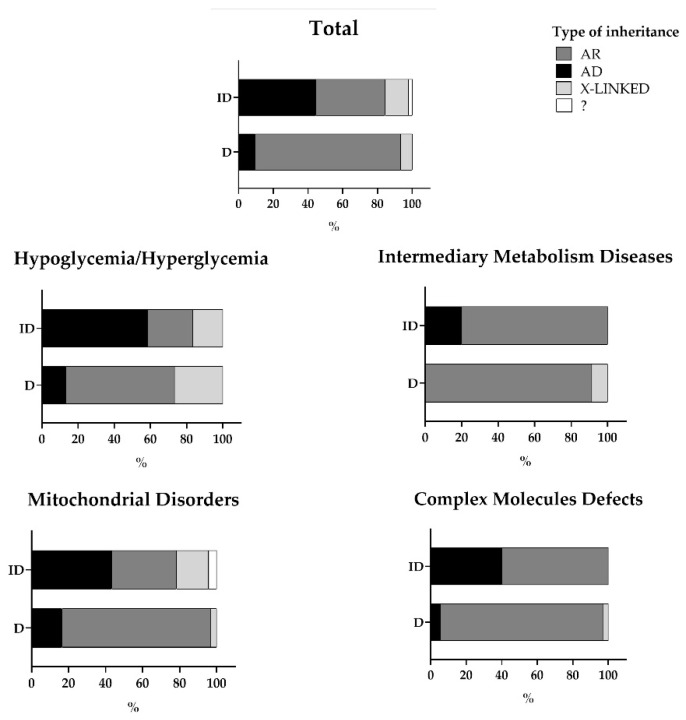
Inheritance pattern for single-gene diseases that were included in each individual panel and identified in successfully and inconclusively diagnosed patients. Abbreviations: ID, inconclusive diagnosis; D, diagnosis; AR, autosomal recessive; AD, autosomal dominant; ?, no inheritance pattern identified in the genes included in the genetic analysis.

**Figure 4 genes-12-01262-f004:**
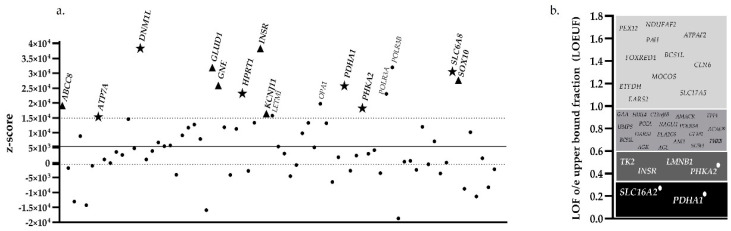
(**a**) Representation of z-scores for autosomal dominant (▲) and X-linked (★) genes that are predicted to be more intolerant to functional variation. (**b**) LOEUF scores from gnomAD, and haploinsufficiency (HI) scores from ClinGen data. Low LOEUF scores (<0.35) indicate strong selection against predicted loss-of-function (pLoF) variation in a given gene. Genes indicated with a white circle have the highest score (3), meaning there is sufficient evidence for haploinsufficiency according to ClinGen.

**Table 1 genes-12-01262-t001:** Characteristics of the study population (patients’ mean age and sex) and technical characteristics of the panels included in the study.

Panel	Age (M ± SD) Years	Sex (%)		Technical Characteristics of Panel	
		Female	Male	Median Coverage (X)	% ≥20X
INT MET	2.00 ± 1.26	34.3	65.7	413.81	99.93
HYPO/HYPER	5.72 ± 6.33	33.9	66.1	360.71	99.17
MITO	6.88 ± 5.57	41.8	58.2	334.05	98.79
COMP MOL	5.38 ± 4.99	44.8	55.2	382.67	97.86
NeuroSeq	5.32 ± 4.61	31.70	68.29	291.50	97.29
Total	6.71 ± 5.73	33.9	60.1	-	-

Abbreviations: COMP MOL, complex molecular defects (including leukodystrophies) panel; HYPO/HYPER, hypoglycemic/hyperglycemic events panel; INT MET, inborn errors of intermediary metabolism panel; M, median; MITO, mitochondrial diseases panel.

## Data Availability

Not available.

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
