# Peer review of "Utility of Gene Panels for the Diagnosis of Inborn Errors of Metabolism in a Metabolic Reference Center"

_genes, 2021, doi:10.3390/genes12081262_

Round 1

Reviewer 1 Report

Overall

The paper describes the diagnostic yield of dedicated metabolic gene panels from a single center.

Data may be worth sharing, but at present the data interpretation and context provided to allow the reader to understand the data needs improvement. This is important to prevent unwarranted conclusions.

Major issues:

Panels should be stably defined. Changes over time of the definition of the panel for this study should be addressed by not including diagnoses made from the added genes, or by discussing these separately.

Two axes are important:

-Was the genetic analysis confirmatory – based on a clear metabolic signature?

-in case of treatable disorders: was a timely diagnosis made based on genetic analysis.

Without these axes the frequency data are essentially uninterpretable because of the influence of these confounders.

Additional comments:

51 ‘To date, there is little evidence supporting genotype–phenotype correlations in IEMs.’ this is rather broadly stated, perhaps false, and at the least requires evidence= references. It is not very relevant for the ‘story line’.

.. panels have been changed along the way. This makes it impossible to have a real percentage.

An option would be to group the results depending on the number of genes tested in panels.

104 What is meant with ‘in coordination with’? Were patients from the 20 other centers included?

132 and further please provide the genes in these panels, as suppl material

195-197 this appears to be a comment how to write a paper rather than a paper.

271 refers to fig 4a à is absent, should be 3a.

Significance between panels mentioned in the first paragraph of the discussion is not the main finding. As mentioned by the authors, the differences reflect varying aims of implementing the panel: High rates in case of a Confirmatory assay, lower rates in case of semi-untargeted analysis.

The size of panels is not taken into account when describing the detection rate. Needs discussing.

Changes in panel content are not discussed.

301 .. becoming the first line diagnostic test… à discuss pros and cons.

Second paragraph ( 305- 338) Low diagnostic rate: suggests that panel is ‘unfinished’.

429 rapid enough? What would be the preferred route? -- discuss Implications for treatment

Author Response

Reviewer 1 - Comments and Suggestions for Authors

The paper describes the diagnostic yield of dedicated metabolic gene panels from a single center. Data may be worth sharing, but at present the data interpretation and context provided to allow the reader to understand the data needs improvement. This is important to prevent unwarranted conclusions.

We really appreciate the reviewer suggestions. It had substantial improved the interpretation of the results obtained in this work.

Major issues:

Panels should be stably defined. Changes over time of the definition of the panel for this study should be addressed by not including diagnoses made from the added genes, or by discussing these separately. Two axes are important:

-Was the genetic analysis confirmatory – based on a clear metabolic signature?

-In case of treatable disorders: was a timely diagnosis made based on genetic analysis.

Without these axes the frequency data are essentially uninterpretable because of the influence of these confounders.

Thank you for giving a very good contribution to the manuscript. For this reason, we have updated the following:

In the Material and Methods section, subheading Study Population we have included: ‘. For each patient we evaluated the following variables: sex, age of symptom onset, clinical signs and symptoms, laboratory test parameters, family history, consanguinity data, genetic analysis of patients and parents, and treatments administered (nutritional and/or pharmacological).’  (lines 110-114)

In the Result section we have included the following: ‘In 53 patients (63.86%) with a clear metabolic profile, we were able to achieve a confirmatory genetic analysis.’ (lines 224-225) ‘We evaluated the impact of genetic diagnosis on medical management (i.e. changes in medication and/or diet, redirection of patient’s care), and found that establishing a genetic diagnosis directly affected medical management in 32 patients (Supplementary Tables S1-S4), and led to initiation of palliative care in other 8 patients. Moreover, all patients and families benefited from timely genetic counseling’ (lines 227-231).

In the Discussion section, we have reformulated the following paragraph: ‘Indeed, in patients with a clear metabolic profile, we were able to achieve a confirmatory genetic analysis.’ (lines 449-450). ‘Establishing a genetic diagnosis led to changes in medical management or the initiation of palliative care in 40/83 (48.19%) patients.’ (lines 454-456).

Additional comments:

  1. Line 51 ‘To date, there is little evidence supporting genotype–phenotype correlations in IEMs.’ this is rather broadly stated, perhaps false, and at the least requires evidence= references. It is not very relevant for the ‘story line’.

According to the reviewer suggestion we have added the reference of Argmann at al, 2016. In this article, the authors refer “The lack of genotype to phenotype correlation greatly impacts the ability to predict a patient’s disease course.” In our manuscript, we have reformulated the sentence for better understanding for the reader: “Evidence supporting genotype–phenotype correlations in IEMs is not always clear.” (lines 52-53)

  1. .. panels have been changed along the way. This makes it impossible to have a real percentage. An option would be to group the results depending on the number of genes tested in panels.

According to the reviewer recommendation we have added figure 1 which represents the positive diagnostic rate according to the number of genes included in the design of the several versions of the gene panels. We have grouped them along the 3 years of study. ‘Several versions of each multi-gene panels were designed during the 3-year study period (Fig. 1). An exponential increase in the rate of positive diagnosis was observed for the COM MOL panel as new genes were added. In the case of the INT MET, HYPO/HYPER, and MITO panels the rate of positive diagnosis remained relatively stable over the last few versions.’ (lines 239-243)

  1. 104 What is meant with ‘in coordination with’? Were patients from the 20 other centers included?

We have reformulated this affirmation according to reviewer suggestion: ‘This prospective multicenter study was conducted out over a 3-year period in the Metabolic Unit of the University Clinical Hospital of Santiago de Compostela (Spain). Patients included were from 20 centers in Spain and Portugal.’ (lines 104-106)

  1. 132 and further please provide the genes in these panels, as suppl material.

According to the reviewer suggestion, gene content for each final version of the gene panel is provided in the Supplementary material, Table S7. ‘The genes included in the latest version of each gene panel are shown in Table S7’ (line 133)

  1. 195-197 this appears to be a comment how to write a paper rather than a paper.

By mistake the sentence was not deleted from the .docx file template. This was corrected, thank you for noticing.

  1. 271 refers to fig 4a à is absent, should be 3a.

You are right. However, as Figure 1 was added to the manuscript, Fig 4a was kept.

  1. Significance between panels mentioned in the first paragraph of the discussion is not the main finding. As mentioned by the authors, the differences reflect varying aims of implementing the panel: High rates in case of a Confirmatory assay, lower rates in case of semi-untargeted analysis.

According to the reviewer suggestion, we have added the following information in Discussion: ‘In this study, we detected differences in the rate of positive diagnosis between gene panels: higher rates were observed in cases for which a previous confirmatory assay was available, as for the INT MET panel, while lower rates were observed in cases of semi-untargeted analysis (e.g. the HYPO/HYPER and MITO panels).’ (lines 326-329)

  1. The size of panels is not taken into account when describing the detection rate. Needs discussing.

According to the reviewer suggestion, we have included the following in the discussion: ‘Comparison of the rate of positive diagnosis across successive versions of each multi-gene panel design showed that at a certain point the addition of genes did not translate to an increase in the rate of positive diagnosis. For example, the rate of positive diagnosis remained relatively stable over the last few versions of the INT MET, HYPO/HYPER, and MITO panels. There are two potential explanations for this observation. Despite recent advances in our knowledge of the molecular bases of IEMs and the discovery of disease-associated genes, (a) the analysis of NGS data from gene panels remains limited to coding regions of the genome, excluding intronic variants, meaning that we are still missing the same regions; or (b) we are failing to take into account the complex gene regulation processes that may underlie the disease, including regulation in cis by promoters, enhancers, and repressors and regulation in trans by transcription factors or microRNAs.’ (lines 332-344)

  1. Changes in panel content are not discussed.

According to the reviewer suggestion, we have updated the discussion as previously said in lines 332-344.

  1. 301 .. becoming the first line diagnostic test… à discuss pros and cons.

According to the reviewer suggestion, we have added the following information to the discussion:  ’On the one hand patients can benefit from genetic testing to obtain clarity about their dis-ease prognosis, prevent the development of more severe forms, and avail of timely genetic counseling for their families. On the other hand, inconclusive genetic results can lead to stress and anxiety, while definitive results indicating that a patient is at increased risk of developing an incurable disease can be disappointing. In our cohort, all patients were being monitored by metabolic disease units specialized in the diagnosis and follow-up of this type of disorder, which greatly facilitated initial orientation of the suspected diagnosis.’ (lines 352-359)

  1. Second paragraph (305- 338) Low diagnostic rate: suggests that panel is ‘unfinished’.

We have reformulated the paragraph for better understanding: ‘In these cases, the HYPO/HYPER panel is more challenging and difficult to design, and it is therefore highly likely that the causative gene may not have been included. Therefore, analysis by CES or WES is advisable to improve the rate of diagnosis when studying patients with hypo/hyperglycemia who present with clinically significant events.’ (lines 363-366)

  1. 429 rapid enough? What would be the preferred route? -- discuss Implications for treatment

According to the reviewer suggestion, treatment management was included in the discussion (previously answered in Major Comments): Result section, lines 227-231 and Discussion section, lines 454-456.

Reviewer 2 Report

Thank you for the opportunity to review the manuscript “Utility of gene panels for the diagnosis of inborn errors of metabolism in a metabolic reference center” by Sofia Barbosa-Gouveia et al.

In my opinion this prospective multicenter study is a very relevant research. Please find my minor comments and suggestions for the Authors:

  1. Please explain the acronyms in the first affiliation: CIBERER, MetabERN.
  2. Line 104: “20 other centers in Spain and Portugal”- please report which centres (i.e. in Acknowledgments).
  3. Line 114: “the Phenomizer tool allowed us to correlate the phenotype with the obtained molecular diagnosis (compbio.charite.de/phenomizer/) [20].” As using this tool is important part of the study it should be explained in more detail.
  4. Line 119: “hypoglycemic/hyperglycemic events associated with metabolic disorders or other processes”- it should be clarify which processes.
  5. Line 127: “the NeuroSeq panel”- please give details regarding this panel (which genes were included).
  6. Line 193: Results “This section may be divided by subheadings. It should provide a concise and precise description of the experimental results, their interpretation, as well as the experimental conclusions that can be drawn.”- was it put in the main text by mistake?
  7. 1. . Diagnostic rates
  • Could you add the information if/how many of the patients have had the biochemical diagnosis before the genetic results?
  • Despite the supplementary materials regarding individual patients it would be valuable for readers to summarise which IEM were most commonly detected, i.e. in a Table.
  1. Line 238: “Of the 112 patients for whom negative results were obtained using the MITO panel, 41 were reanalyzed upon request using the NeuroSeq panel.”- how were these patients chosen for the widening the diagnostic process?
  2. Line 135 and following: I would recommend clarifying how many patients in each group were diagnosed with “the first designed version” vs further versions.
  3. Line 170: “To ensure a reliable clinical interpretation of the variants detected, we applied prioritization criteria to predict pathogenicity according to AMCG guidelines [29].”- please explain the guidelines in more detail.
  4. Line 265: “Comparing the expected and the observed number of rare missense variants in population databases, such as gnomAD, can provide information on pathogenicity if a gene demonstrates constraint for missense variation, depending on whether that constraint is observed across the entire gene or is confined to specific regions of the gene.”- this part of the text is unclear, please revise the meaning of the paragraph and add the information about “gnomAD”.
  5. Line 271: “we detected missense variants in intolerant genes”; line 377: “We found that statistically significant data were useful to help prioritize likely pathogenic variants in genes expected to be intolerant to mutation.” As the paper is important not only for the geneticists but as well for the clinicians, I would recommend adding a concise explanation of the term “intolerant genes”.
  6. Line 458: “informed consent”- it should be “written informed consent”.

Author Response

Reviewer 2 - Comments and Suggestions for Authors

Thank you for the opportunity to review the manuscript “Utility of gene panels for the diagnosis of inborn errors of metabolism in a metabolic reference center” by Sofia Barbosa-Gouveia et al. In my opinion this prospective multicenter study is a very relevant research. Please find my minor comments and suggestions for the Authors:

We really appreciate the reviewer suggestions. It had improved the interpretation of the results obtained in this work.

  1. Please explain the acronyms in the first affiliation: CIBERER, MetabERN.

According to the reviewer suggestion, the acronyms were explained (lines 8 and 9)

2. Line 104: “20 other centers in Spain and Portugal”- please report which centres (i.e. in Acknowledgments).

According to the reviewer suggestion, we have included the 20 centers who participated in this study in Acknowledgments. (lines 524-534)

3. Line 114: “the Phenomizer tool allowed us to correlate the phenotype with the obtained molecular diagnosis (compbio.charite.de/phenomizer/) [20].” As using this tool is important part of the study it should be explained in more detail.

According to the reviewer suggestion we have added the following information: ‘This computer-based tool can facilitate the diagnostic process by combining specific phenotypic features, correlating them with a genetic disorder, and generating a corresponding p-value.’ (lines 122-124)

4. Line 119: “hypoglycemic/hyperglycemic events associated with metabolic disorders or other processes”- it should be clarify which processes.

According to the reviewer suggestion we have added the following information: (i.e. Alström Syndrome, Shashi-Pena syndrome, tubulointerstitial kidney disease).’ (lines 151-152)

5. Line 127: “the NeuroSeq panel”- please give details regarding this panel (which genes were included).

According to the reviewer suggestion, gene content for each final version of the gene panel is provided in the Supplementary material, Table S7. ‘The genes included in the latest version of each gene panel are shown in Table S7’ (line 133)

6. Line 193: Results “This section may be divided by subheadings. It should provide a concise and precise description of the experimental results, their interpretation, as well as the experimental conclusions that can be drawn.”- was it put in the main text by mistake?

By mistake the sentence was not deleted from the .docx file template. This was corrected.

7. Diagnostic rates

  • Could you add the information if/how many of the patients have had the biochemical diagnosis before the genetic results?

According to the reviewer suggestion, we have included the following in the Result section: ‘In 53 patients (63.86%) with a clear metabolic profile, we were able to achieve a confirmatory genetic analysis. (lines 225-226).

  • Despite the supplementary materials regarding individual patients it would be valuable for readers to summarise which IEM were )most commonly detected, i.e. in a Table.

According to the reviewer suggestion, we have included the following in the Result section:

‘The most common IEMs detected in this study were combined oxidative phosphorylation deficiency (11 patients),) phenylketonuria (7 patients) and   lysosomal disorders (7 patients).’ (lines 261-263)

8. Line 238: “Of the 112 patients for whom negative results were obtained using the MITO panel, 41 were reanalyzed upon request using the NeuroSeq panel.”- how were these patients chosen for the widening the diagnostic process?

When NeuroSeq panel was first designed, we informed the physicians about reanalyzing patients with a negative molecular result. Those patients who agreed to participate (41) where reanalyzed.

9. Line 135 and following: I would recommend clarifying how many patients in each group were diagnosed with “the first designed version” vs further versions.

According to the reviewer suggestion, we have added figure 1 which represents the positive diagnostic rate according to the number of genes included in the design of the several versions of the gene panels. We have grouped them along the 3 years of study. ‘Several versions of each multi-gene panels were designed during the 3-year study period (Fig. 1). An exponential increase in the rate of positive diagnosis was observed for the COM MOL panel as new genes were added. In the case of the INT MET, and HYPO/HYPER panels the rate of positive diagnosis remained relatively stable over the last few versions.’ (lines 240-244)

10. Line 170: “To ensure a reliable clinical interpretation of the variants detected, we applied prioritization criteria to predict pathogenicity according to AMCG guidelines [29].”- please explain the guidelines in more detail.

According to the reviewer suggestion, we have added the following information: ‘Based on this classification system variants are assigned to one of 5 pathogenicity classes: likely benign, benign, uncertain significance, likely pathogenic, and pathogenic’ (lines 184-185)

11. Line 265: “Comparing the expected and the observed number of rare missense variants in population databases, such as gnomAD, can provide information on pathogenicity if a gene demonstrates constraint for missense variation, depending on whether that constraint is observed across the entire gene or is confined to specific regions of the gene.”- this part of the text is unclear, please revise the meaning of the paragraph and add the information about “gnomAD”.

According to the reviewer suggestion, we have reformulated the paragraph: ‘Comparing the expected and the observed number of rare missense variants in population databases, such as gnomAD, (which aggregates exomes and whole genome data from a large groups of individuals and enables identification of common and rare variants), can provide information on pathogenicity if a gene demonstrates constraint for missense variation, depending on whether that constraint is observed across the entire gene or is confined to specific regions of the gene. Larger genes are expected to have more missense variants, although this does not mean the observed variants are pathogenic. Certain genes include highly-conserved regions that are related to the strength of purifying selection and are more sensitive to variation, meaning that rare variants identified in these genes are more likely to be associated with disease.’ (lines 296-305)

12. Line 271: “we detected missense variants in intolerant genes”; line 377: “We found that statistically significant data were useful to help prioritize likely pathogenic variants in genes expected to be intolerant to mutation.” As the paper is important not only for the geneticists but as well for the clinicians, I would recommend adding a concise explanation of the term “intolerant genes”.

According to the reviewer suggestion, we have reformulated the paragraph: ‘These measures were used to determine how intolerant a gene can be to a specific type of variant. Intolerant genes are those that are more likely to cause disorders than genes that can tolerate functional variation.’ (lines 191-192)

13. Line 458: “informed consent”- it should be “written informed consent”.

According to the reviewer suggestion, this was corrected.

Reviewer 3 Report

To the authors,

The article “Utility of gene panels for the diagnosis of inborn errors of metabolism in a metabolic reference center” is a clear and well-structured manuscript. However, I have some question and change suggestions.

  1. In the abstract you don´t specify that the rate of positive diagnosis are per each panel, and it can lead to confusion. Please re-write it.
  2. When you discuss about the patient 38, you say that the p-value obtained with Phenomizer is significant. However, I do not obtain that result when I introduce the phenotype characteristics in the program. I have obtained a p value = 1 and the gene is ranked at the position 830. Could you explain discrepancy?

There are some minor changes, which should be corrected:

  1. The first three lines of the Result section should be remove (lines 194 to 196).
  2. Rename table 2 to table 1 in line 200.
  3. Before the subheadings of the Result section there is a “dot” which should be removed (lines 209, 250 and 263)
  4. Do the lines 235 to 237 belong to the legend of the figure 1? If not, I suggest put them there.
  5. Rename Fig.4A to Fig.3A in line 270
  6. When you name the genes (ATP7A, HPRET1, PHKA2, etc) they should be in italics. Some examples of this mistake are found in lines 273 to 275, 277, 278, 329, 337, 344 to 345, 352 and 384.
  7. Rename PDH1 to PDHA1 in line 384.

Best regards,

Author Response

Reviewer 3 - Comments and Suggestions for Authors

To the authors: The article “Utility of gene panels for the diagnosis of inborn errors of metabolism in a metabolic reference center” is a clear and well-structured manuscript. However, I have some question and change suggestions.

We really appreciate the reviewer suggestions. It had substantial improved the interpretation of the results obtained in this work.

  1. In the abstract you don´t specify that the rate of positive diagnosis are per each panel, and it can lead to confusion. Please re-write it.

Information included in Abstract: ‘The rate of positive diagnosis was 61.86% for intermediary metabolism defects, 32.84% for complex molecular defects, 19% for hypoglycemic/hyperglycemic events, and 17% for mitochondrial diseases,’ (lines 24-26)

2. When you discuss about the patient 38, you say that the p-value obtained with Phenomizer is significant. However, I do not obtain that result when I introduce the phenotype characteristics in the program. I have obtained a p value = 1 and the gene is ranked at the position 830. Could you explain discrepancy?

It was a mistake by using the data from the original database. Thank you very much for your detection. We have corrected the phenotypic features for that patient and the corresponding p-value. Thanks to your comment we have reviewed again all the patients included in the table and other 2 discrepancies were identified and corrected.

There are some minor changes, which should be corrected:

  1. The first three lines of the Result section should be remove (lines 194 to 196).

By mistake the sentence was not deleted from the .docx file template. This was corrected.

2. Rename table 2 to table 1 in line 200.

According to the reviewer suggestion, we have renamed Table 2 to Table 1. (line 213)

3. Before the subheadings of the Result section there is a “dot” which should be removed (lines 209, 250 and 263)

According to the reviewer suggestion, we have removed the “dot” before the subheadings, in the Results section.

4. Do the lines 235 to 237 belong to the legend of the figure 1? If not, I suggest put them there.

Lines 235 to 237 indeed belong to the legend of the figure, this was corrected. This is now figure 2, lines 265-267.

5. Rename Fig.4A to Fig.3A in line 270

You are right. However, as Figure 1 was added to the manuscript, Fig 4a was kept.

6. When you name the genes (ATP7A, HPRET1, PHKA2, etc) they should be in italics. Some examples of this mistake are found in lines 273 to 275, 277, 278, 329, 337, 344 to 345, 352 and 384.

You are right. We have corrected the name of the genes according to the nomenclature recommendations and the reviewer suggestion.

7. Rename PDH1 to PDHA1 in line 384.

You are right. According to the reviewer suggestion, we have corrected PDH1 to PDHA1.
